

# Design of sports training information analysis system based on a multi-target visual model under sensor-scale spatial transformation

Mei Hu, Ming Zhang and Kewei Yu

Physical Education Department, Guangzhou Xinhua University, Guangzhou, China

Corresponding author
Kewei Yu, bcykw@xhsysu.edu.cn

## ABSTRACT

In the contemporary realm of athletic training, integrating technology is a pivotal determinant for augmenting athlete performance and refining training outcomes. The amalgamation of multi-target visual modeling with sensor technology imparts an enriched stratum of sports training data. Subsequently, the sensor scale-space transformation accentuates the comprehensive apprehension of data across diverse scales and angles. Hence, within this manuscript, addressing the multi-target tracking intricacies during sports training and competition, we posit a framework that amalgamates the shortest path elucidated by the K shortest paths (KSP) methodology with the pose information emanating from the Alphapose network. This framework recognizes the athlete's shortest path through a convolutional neural network and KSP, followed by the amalgamation of these divergent data sources. The fusion unfolds by incorporating the athlete's pose information grounded in Alphapose, culminating in a comprehensive integration of the two data streams. Consequently, synthesizing alpha-derived athlete information precipitates the ultimate amalgamation of the two information streams. The accomplished fusion, premised on Alphapose, forms the bedrock for multi-target tracking, culminating in a feature-rich synthesis. Empirical results reveal that after integrating these information streams, the Multiple Object Tracking Accuracy (MOTA) index and Global Multiple Object Tracking Accuracy (GMOTA) index surpass those of the solitary information tracking methods, thereby furnishing a technical underpinning and a foundation for information fusion within prospective sports training analysis systems.

## INTRODUCTION

In the era of burgeoning network technology and heightened societal attention towards well-being, sports videos assume an increasingly paramount role in our daily existence. This burgeoning trend not only permeates the realm of entertainment but also serves as a linchpin in propelling the evolution of sports. Proficiently scrutinizing and comprehending these sports videos enhances the audience's viewing experience and furnishes athletes and coaches with profound insights during training and competitions, catalyzing innovation and fortification across the sports industry. The discernment and estimation of athletes'

presence and poses in sports videos emerge as the bedrock for attaining comprehensive analysis and comprehension of video content (*Wang & Du, 2022*). By leveraging deep learning methodologies, athletes' motions and stances can be discerned and encapsulated with remarkable precision, empowering coaches to scrutinize technical nuances with heightened granularity. Consequently, this facilitates the provision of more meticulous guidance and training counsel. This personalized analysis proves advantageous not solely to professional athletes but also extends its benefits to amateurs, fostering refinement in their sports proficiency and engendering heightened participation in sports among the broader populace (*Nagai, Akashi & Sugino, 2022*).

In sports video analysis and comprehension, the salient role of multi-target visual tracking technology becomes indisputable. This technological facet facilitates precise tracking of multiple athletes or objects within a video, thereby endowing sports analysis with a more expansive and profound dataset. Among its applications, athlete trajectory analysis stands out, enabling the ability to trace athletes' movement paths on the field. It equips coaches with pivotal insights into movement patterns and strategic applications (*Zou et al., 2023*). Furthermore, the team's tactical analysis, real-time sports condition monitoring, and competition data analysis derive significant advantages from integrating multi-target visual tracking technology. The deployment of this technology not only amplifies the real-time surveillance of games and training sessions but also furnishes robust support for the burgeoning sports industry and the in-depth assessment of athletes' performances. The deep learning models for target detection and tracking, exemplified by You Only Look Once (YOLO) and Faster recombinant convolutional neural network (R-CNN), exhibit adeptness in inefficient target recognition; in parallel, conventional multi-target tracking algorithms like simple online and realtime tracking (SORT) and multiple object tracking (MOT) execute target association through the amalgamation of motion models and appearance features. Complementary methodologies, such as multi-camera cooperative tracking and deep association networks, are pivotal in refining accuracy and fortifying robustness (*Kaur & Singh, 2022*). Through multi-target tracking and motion analysis, we can better analyze the state of athletes and evaluate their performance on the field through trajectory analysis of target objects. However, despite the mature development of deep learning object detection methods, the selection of object detection methods still needs to be investigated due to the difficulty of camera placement and related tasks.

In the realm of multi-target tracking, conventional Kalman filtering and extended Kalman filtering exhibit commendable performance in addressing linear systems, characterized by the merits of low computational overhead and real-time efficiency, rendering them suitable for scenarios governed by simplistic motion models. Conversely, particle filtering methods, harnessing random sampling, adeptly navigate nonlinear and non-Gaussian motion models, showcasing heightened flexibility in the face of target motion uncertainty and applicability to intricate tracking scenarios. Correlation filter methods leverage the Fourier domain for target detection and tracking, boasting efficient computational prowess and compatibility with scenarios demanding both natural high real-time capability and precision (*Balakrishna & Mustapha, 2023*). Deep learning methods such as convolutional neural networks (CNN) and recurrent neural networks (RNN) have

achieved significant success in object detection and tracking, adapting to more complex scenarios by learning the semantic features and motion patterns of targets. Graph-based methods can effectively model the correlation between targets.

In contrast, trajectory graph optimization methods improve the accuracy of target trajectories through globally consistent trajectory optimization, making them particularly suitable for long-term target tracking (*Wang et al., 2023*). On this basis, introducing the K shortest paths (KSP) algorithm further optimizes the dynamic correlation between targets by finding multiple shortest paths, significantly enhancing the flexibility and robustness of multi-target tracking. Applying the KSP algorithm deals with occlusion and complex interactive scenes. It improves tracking accuracy and reliability by considering multiple possible trajectories, providing a powerful tool for multi-target tracking in complex environments. Hence, a judicious amalgamation of contemporary deep learning techniques, integrating image information and fusing target detection with positional data, emerges as the optimal strategy for augmenting the efficacy of target detection. In this manuscript, we posit a network framework predicated on the shortest path KSP method and the fusion of posture information, orchestrating an optimal solution for the multi-target tracking problem inherent in sports training. The distinctive contributions of this work are delineated as follows:

1. Leveraging sports video data, the KSP method is employed for the shortest path recognition of athletes, culminating in the realization of path recognition throughout the target tracking process.

2. Employing Alphapose, extracting athletes' postural features enhances information usability—subsequently, posture and path information fusion consummated at the decision-making level.

3. Grounded in the combined path and attitude information, the high-precision tracking of athletes in sports videos is actualized. Rigorous testing under diverse backgrounds substantiates the significant enhancement in target tracking performance after information fusion.

The rest of the article is organized as follows: 'Related works' introduces the related work for object detection and multi-target tracking. 'Methodology' establishes the proposed framework. 'Experiment results and analysis' gives the experiment details and results, and the conclusion is drawn at the end.

## RELATED WORKS

Considering the demand for video information training and analysis in sports systems, it mainly includes two processes. Firstly, it is necessary to perform target detection on athletes, that is, confirm the current target and then perform motion analysis on the detected target to better understand athletes' training and competition status. Therefore, in this article, we first analyze the current situation of object detection and then analyze its detailed application in sports training.

## Target detection

Target detection is a foundational pursuit within computer vision, garnering substantial attention in research endeavors. Before the advent of deep neural networks, target detection algorithms primarily relied on extracting manual image features for subsequent feature matching. Esteemed manual features such as SIFT and SURF yielded commendable results. Many initiatives sought to tailor diverse manual features for target detection, aligning with the specific characteristics of the target; examples include V-J detection (*Viola & Jones, 2001*), HOG detection (*Dalal & Triggs, 2005*) algorithms, and the like. In recent years, propelled by the evolution of deep learning, target detection algorithms founded on convolutional neural networks have ascended to prominence. Concurrently, the introduction of large-scale universal target detection datasets has furnished indispensable data support for convolutional neural network-based target detection algorithms. Many efforts have delved into diverse convolutional neural network structures to realize effective target detection. The YOLO series (*Redmon & Farhadi, 2017*) perceives detection as a regression problem, seamlessly predicting target categories and locations end-to-end through convolutional neural networks. The Faster-RCNN series (*Girshick, 2015*) enhances detection accuracy by bifurcating target detection into two phases: initially extracting foreground frames potentially containing the target, followed by detailed classification and regression. *Lin et al. (2017)* optimize the detection performance of densely distributed small targets by introducing focal loss. *Xiao & Jae Lee (2018)* elevate video detection performance by incorporating a sequence of spatiotemporal memory stores. *Zhu et al. (0000)* harness optical flow to capture motion information in videos, seamlessly integrating it with feature maps in CNN to achieve end-to-end video detection. *Zhang & Wang (2016)* delve into stability in video detection, mitigating detection frame jitter by incorporating a tracking strategy. *Yin & Liu (2017)*, grounded in the concept of multi-cascading, decompose target detection into multiple classification and regression problems, achieving precise face detection. *Liu et al. (2019)* discard the conventional window detection approach, opting to directly predict the body center and dimensions, thereby realizing pedestrian detection.

## Posture estimation for sports movement

As evident from the current state of target detection research, the continuous evolution of deep learning technology has led to the maturation of single-target monitoring research. However, a notable gap exists in sports and training as the research landscape predominantly addresses single-target tracking scenarios. Hence, there is a pressing need to convene seminars addressing the intricacies of multi-target tracking. The primary mission of Multiple Object Tracking (MOT) revolves around identifying the positions of all targets of interest within an image sequence in each frame while preserving their identity information. These targets include people, vehicles, animals, and other moving objects. Notably, pedestrian tracking has been the most explored domain, finding applications in intelligent surveillance and autonomous driving. Yet, the challenges in sports scenarios, marked by dynamic and unpredictable player movements, render multi-target tracking more complex than pedestrian tracking. *Yu et al. (2003)* introduced a ball detection method grounded in trajectory analysis. This method initiates by detecting multiple

candidate targets during the detection phase, followed by trajectory analysis to ascertain the ball's movement trajectory. *Liang et al. (2005)* proposed a comprehensive ball-tracking algorithm that amalgamates detection and tracking. The algorithm tracks the ball *via* Kalman filtering by detecting candidate objects and computing the optimal path using the Viterbi algorithm. It determines whether the tracker has lost the ball by evaluating the tracked region and repeating the detection and tracking process. Acknowledging the challenges of adapting single-camera-based tracking to prolonged ball occlusion, scholars have proffered various multi-camera tracking schemes (*Yoon, Song & Jeon, 2018*). *Ren et al. (2009)* proposed a method based on multiple stationary cameras for football games, leveraging the characteristics of the ball's size, appearance, and moving speed. The approach utilizes the Kalman filtering method to derive the ball trajectory, subsequently fusing trajectory segments from different viewpoints into the 3D trajectory of the ball. To address occlusion challenges within a multi-camera framework, *Wang et al. (2014)* presented a ball-tracking method that considers the dynamic relationship between players and the ball. *Ivankovic et al. (2012)* employed the AdaBoost algorithm for player detection in football and basketball scenarios, utilizing hand-designed directional gradient histogram HOG features and Haar features. While effective for pedestrian detection, these methods exhibit limitations in player detection tasks due to the large posture variations and significant occlusions in sports videos. It is attributed to the interference and reduced effectiveness of traditional manual features in facing such challenges. In recent years, the ascendancy of deep learning has steered the focus toward player detection methods based on convolutional neural networks. For instance, *Lu et al. (2018)* proposed a player detection method addressing issues such as substantial changes in the player body and scale induced by camera movement, applicable across diverse and complex sports scenes, including basketball, football, and ice hockey.

The preceding research underscores that integrating deep learning and diverse target detection methodologies facilitates a nuanced exploration of sports competition or training information within sports training and video data analysis. This approach enables in-depth performance analysis of individual players, imparting significant value to enhancing athletes' skill levels. Notably, using deep convolution, multimetric vision, and other advanced methods in target detection proves highly effective for addressing the challenges associated with multi-target detection tasks. The amalgamation of multidimensional data becomes imperative given the importance of positional status, along with players' nuanced body shapes and clothing information during sports training. In this context, realizing multi-source data fusion is a pivotal strategy for achieving enhanced target detection capabilities.

## METHODOLOGY

After analyzing the current application of deep learning technology in athlete training research, this article intends to integrate athlete's path information and pose information to complete multi-objective tracking tasks and improve the effectiveness of athlete target tracking.

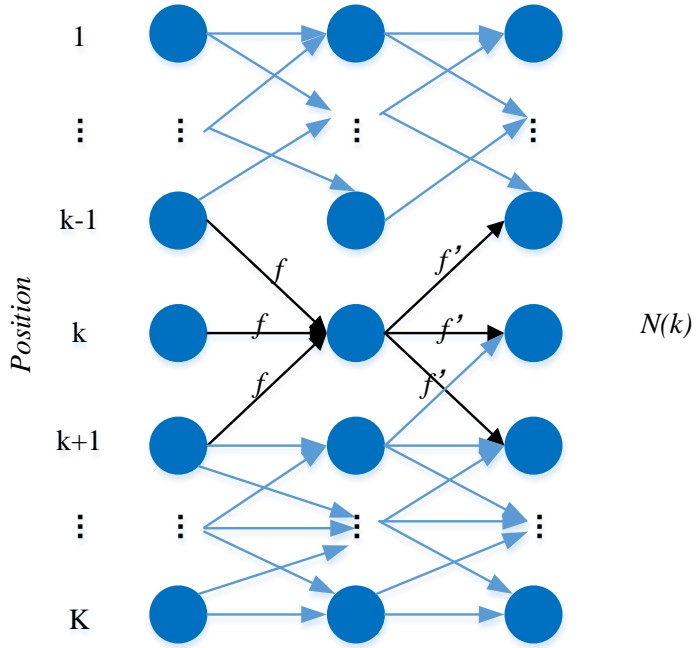

**Figure 1** The one dimensional node model.

## KSP-based multi-target tracking

The k-shortest-path algorithm (*Maidana et al., 2024*), a venerable concept in graph theory, is fundamentally employed to identify the first K paths in a graph representing the shortest routes between specified start and end points. The applicability of the KSP algorithm to addressing the multi-objective tracking problem stems from its capacity to model the situation as a linear programming challenge. That facilitates the construction of a network flow graph based on temporal and spatial parameters, subtly transforming the problem into a graph optimization endeavor to identify the shortest paths within the spatiotemporal graph.

The KSP tracking algorithm holds a distinctive advantage in its simplicity of inputs, encompassing solely the positional coordinate points of tracked targets and their corresponding features. Effectively transforming the multi-target tracking problem into an integer programming challenge, the algorithm operates under the assumption that the current video stream under consideration for tracking comprises T frames. Each frame, in turn, detects K candidate targets, represented as nodes in the algorithm. The inter-frame relationships between candidate targets are established as edges based on distance constraints, culminating in the depiction of a one-dimensional node model graph, as illustrated in Fig. 1.

Each node variable 'n' signifies the number of target objects presently situated at that node, and the weight of each edge indicates the flow of 'f' targets through that particular connection. Building upon this foundation and taking into account the two-dimensional nature of the camera data, we can posit that the probability distribution for the player's

position on the pitch plane is as follows:

$$\rho_i^t = P\left(X_i^t = 1 | I^t\right) \tag{1}$$

where $\rho_i^t$ denotes the position of the course plane, i is the probability that a player is present at frame t, The likelihood that a player is present at frame 1, $X_i^t$ is a random variable, and $X_i^t = 1$ is a random variable, then it means that the player exists at t player exists at the time; otherwise it does not exist, and $I^t$ is the probability that a player exists at the time of frame t frame corresponds to the court plan.

In the realm of multi-target tracking, the overarching objective resides in the identification of numerous tracking trajectories. Facilitating these trajectories' interconnection becomes paramount, optimizing the likelihood of capturing players' movements. Building upon this premise and leveraging the extant probabilities, the challenge of multi-objective tracking transforms into a linear programming paradigm.

$$\max \sum_{i=1}^{T} \sum_{i=1}^{G} \log\left(\frac{\rho_i^t}{1-\rho_i^t}\right) \cdot \sum_{j \in \mathbb{N}(i)} f_{i,j}^t \tag{2}$$

Its constraint is:

$$\forall t, i, f_{i,j \in \mathbb{N}(i)}^t \geq 0 \tag{3}$$

$$\forall t, i, \sum_{j \in \mathbb{N}(i)} f_{i,j}^t \leq 1 \tag{4}$$

$$\forall t, i, \sum_{j \in \mathbb{N}(i)} f_{i,j}^t - \sum_{k \in \mathbb{N}(j)} f_{j,k}^{t-1} \leq 0 \tag{5}$$

where T denotes the number of frames in a batch, G denotes the number of nodes in the network flow graph, and $\mathbb{N}(i)$ represents node I, the neighborhood of a node. J is one of the neighbors of i, one of the neighboring nodes of the node, the $f_{i,j}^t$ is one of the neighboring nodes of t at the moment of flow from node i to the adjacent node j at the time of flow from the node to the neighboring node, and $\rho_i^t$ denotes the position of the pitch plane i at the node in the first t The probability that a player is present at the frame. In addition, the three constraints indicate that the number of network flows is all non-negative, and the number of network flows between two positions is 0 or 1. In this solution process, since the constraint matrix has the property of unimodularity, it can be solved by the above conditions using the relaxation linear programming method and converge to the integer solution quickly.

The decision to employ the KSP methodology and AlphaPose network is underscored by their exceptional accuracy, robustness, scalability, and state-of-the-art performance in human pose estimation tasks. Their combination offers a reliable solution capable of handling diverse environmental conditions, large datasets, and real-time processing requirements. Furthermore, their open-source availability fosters accessibility and

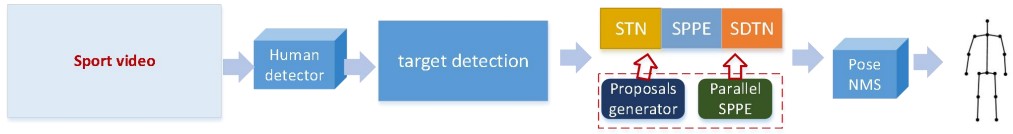

**Figure 2** **The framework for the Alphabet.**

collaboration within the research community, driving further innovation. This selection ensures the success of specific projects and contributes to advancing human-centric technologies. Overall, the choice of the KSP methodology and AlphaPose network reflects a commitment to excellence, innovation, and impact in human pose estimation and beyond.

## Alphabet-based multiplayer attitude estimation

Human pose estimation methodologies can be dichotomously categorized based on the sequencing of their procedural steps: top-down and bottom-up recognition methods. The top-down approach involves initially pinpointing the precise localization of each individual in the image, followed by the subsequent recognition of crucial points for individual persons. This method's critical key point recognition accuracy is often contingent upon the precision of human body localization. Conversely, the bottom-up methodology first detects all vital points in the image, then groups and aggregates these key points to delineate each individual. Although this method exhibits a faster recognition speed, its accuracy tends to be lower than the aforementioned top-down approach. The AlphaPose human pose estimation model employed in this study (*Halleck et al., 2023*) adheres to the fundamental framework of the Regional Multi-Person Pose Estimation (RMPE) proposed by the research team. Notably, this model iteration excels in the real-time estimation of multi-person poses while ensuring commendable recognition outcomes. The comprehensive flow of the entire framework is elucidated in Fig. 2.

AlphaPose harnesses convolutional neural networks (CNNs) to primarily extract image features, employing multiple layers of convolution, activation functions (*e.g.*, ReLU), and pooling layers. This process maps the image into the feature space, facilitating the discernment of human vital points. Each key point's location is determined by generating a corresponding heat map, with localization further refined using local maxima. Initially, the input image undergoes scrutiny by the human detector to delineate the target region of the human body. The YOLOv5 module, renowned for its current popularity and accuracy, is employed within the human detector module. Subsequently, the body region frame is directed SSTN into the SSTN, including the STN, SPPE, and SDTN. Finally, all candidate poses are subjected to the P-NMS (Parametric Pose Non-Maximum-Suppression) module. A traversal of predicted candidate poses is necessary within the P-NMS module (*Sabo et al., 2023*). Poses with confidence scores surpassing a predefined threshold are selected, and the pose with the highest confidence is designated as the reference pose. Subsequently, the elimination criteria are applied, involving the calculation of distances between the remaining poses and the reference pose. Alphapose provides significant advantages in

target tracking through attitude estimation, mainly reflected in its high accuracy and strong robustness. By accurately identifying human key points, Alphapose can accurately track multiple targets in complex environments, maintaining stability even in occlusion or rapid motion. In addition, its sensitive ability to capture dynamic postures provides rich information for understanding target behavior, which is particularly important in application scenarios such as sports training and monitoring. The integrated Alphapose tracking system can enhance the coherence of target recognition and tracking through attitude information, improving overall tracking performance. Therefore, Alphapose enhances the accuracy of multi-target tracking and provides a powerful tool for in-depth analysis of target behavior.

When distances fall below the threshold, the corresponding poses are eliminated. This process iterates until all poses conform to the threshold criteria (*Zwölfer et al., 2023*). The elimination criterion is defined by Eq. (6):

$$f\left(P_i, P_j | \lambda, \eta\right) = 1\left(d\left(P_i, P_j | \lambda, \lambda\right) \leq \eta\right) \tag{6}$$

where $d()$ calculates the spatial distance of the two poses and the weighted distance of the pose distances. If the distance is less than or equal to the threshold $\eta$, then it represents the reference gesture $P_i$ and the pose $P_j$ are too similar, and the gestures need to be eliminated. The formula for calculating the distance which is shown in Eq. (7):

$$d\left(P_i, P_j | \lambda\right) = K_{Sim}\left(P_i, P_j | \sigma_1\right) + \lambda H_{Sim}\left(P_i, P_j | \sigma_2\right) \tag{7}$$

where $\lambda = \{\sigma_1, \sigma_2, \lambda\}$, $K_{Sim}$ represents the stance distance, $H_{sim}$ represents the spatial distance. Assuming that $Bi$ is the predicted bounding box of $Pi$, we can compute the pose distance by using Eq. (8):

$$K_{im}\left(P_i, P_j | \sigma_1\right) = \begin{cases} \sum_n \tanh \frac{c_i^n}{\sigma_1} \tanh \frac{c_j^n}{\sigma_1} & \text{if } k_j^n \text{ is within } B\left(k_i^n\right) \\ 0 & \text{otherwise} \end{cases} \tag{8}$$

The spatial distance is used to calculate the spatial similarity of the distance of the corresponding feature information between two poses, which can be related by Eq. (9).

$$H_{Sim}\left(P_i, P_j | \sigma_2\right) = \sum_n \exp\left[-\frac{\left(k_i^n - k_j^n\right)^2}{\sigma_2}\right]. \tag{9}$$

## Object tracking framework fusing KSP and position estimation information

Upon concluding the extraction of attitude information and shortest path details in the multi-target tracking process, we operationalize the multi-target tracking objective within the sports training regimen. That is achieved by utilizing both the shortest path information and position constraint data.

The framework begins by utilizing YOLO to delineate target divisions accurately. Subsequently, AlphaPose is employed to compute the joints of all individuals in the

diagram. Joint matching is conducted, resulting in a diverse array of human body postures derived from the joints. Athlete posture matching is executed based on the athlete's detection frame, eliminating extraneous human body postures. This process ultimately yields the posture estimation of athletes within the diagram.

Adopting a bottom-up approach, which returns to the joints of the entire graph before assembling different human postures, mitigates the issue of missed detections stemming from people aggregation. This approach also obviates the need for repetitive calls to the single-person posture detector.

Simultaneously, optimal path selection is achieved through the KSP method based on recognized targets. Features such as jersey and player number are extracted using the ReID network (*Zhong et al., 2018*), known for its prowess in discerning the same individual across diverse scenes. The ReID network's probability is computed through the Probability of Model (POM) method, prioritizing consideration of model classification error probability over mere accuracy. Inputs for the KSP method are derived from these features, followed by subsequent shortest path computation and heatmap generation. Finally, the generated shortest path information and bitmap details are fused at the decision level, creating new feature vectors that effectively fulfill the multi-target tracking objective.

The convergence process seamlessly integrates data from the CNN and the KSP method before incorporating alpha-derived data. Initially, the CNN extracts features such as jersey and player number, while the KSP method facilitates optimal path selection based on recognized targets. These features converge as inputs for subsequent shortest-path computation, ensuring paths encapsulate spatial relationships and target-oriented trajectories. Following this convergence, Alphapose-derived data is seamlessly integrated. AlphaPose network computes the joints of all individuals, providing crucial pose information. This data enriches generated paths with context for human movements (*Fleuret et al., 2007*). By integrating Alphapose-derived pose information with CNN and KSP-established paths, the convergence process achieves a comprehensive understanding of the ethene, accounting for spatial and kinematic aspects of human activity. This transition ensures a holistic approach to multi-target tracking, leveraging each component's strengths to enhance accuracy and reliability. The smooth integration of CNN, KSP, and Alphapose data streamlines the convergence process, improving understanding and application across various domains, from surveillance to sports analytics.

Integrating pose information with shortest path computation early in the framework development establishes a vital connection between these components, enhancing overall performance. This integration ensures paths are optimal and contextually relevant to human postures, improving tracking accuracy. Considering the spatial constraints imposed by poses, the system generates paths aligning closely with actual movements, reducing false positives and erroneous paths.

# EXPERIMENT RESULTS AND ANALYSIS

Following the establishment of the model, we meticulously curated pertinent datasets for model validation. A prudent data elimination process was executed during dataset

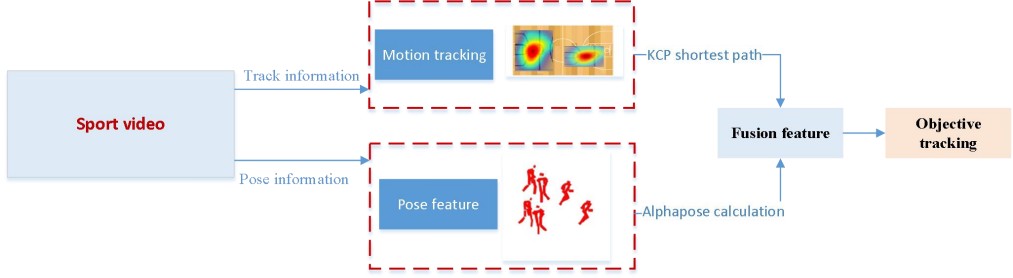

**Figure 3** The framework for the proposed Multiple Object Tracking system for physical training.

**Table 1** The specific information for the employed dataset.

| Dataset | Information |
| --- | --- |
| APIDIS (*Fleuret et al., 2007*) | Simultaneous acquisition by seven cameras, five of which are mounted on the pitch surface and two of which are fisheye cameras mounted on the top of the pitch. |
| Volleyball (*De Vleeschouwer et al., 2008*) | A total of 55 different volleyball videos were collected in the dataset, and 4,830 of these video frames were annotated with athletes with more than 50,000 edges. |
| SPIROUDOME (*Ibrahim et al., 2016*) | Unlike common broadcast tournaments, it is fixed-view basketball surveillance video. The dataset provides rich edge labelling for supervised training. |

selection, model training, and subsequent data selection and experimentation phases to align with the algorithm's practical applicability. Subsequent subsections will expound upon comprehensive details regarding the evaluation indices used in our experiments.

## Experiment setup

Upon the completion of the model construction, as depicted in Fig. 3, the ensuing phase involves validating the model across diverse datasets. This study focuses on data sourced from three distinct sports for comprehensive data analysis. The datasets originate from videos captured during sports events and institutions associated with sports training. Specific details about these datasets are elucidated in Table 1.

Upon finalizing the selection and construction of pertinent data, our objective shifts toward assessing and comparing the model's efficacy. For this purpose, we opt for commonly employed evaluation metrics in multi-target tracking. In this article, our evaluation hinges on multiple object tracking accuracy (MOTA) and global multiple object tracking accuracy (GMOTA), renowned metrics for gauging the accuracy and effectiveness of algorithms in detecting and tracking multiple objects within video sequences. The computation of MOTA is delineated by Eq. (10):

$$\text{MOTA} = 1 - \frac{\sum_t \left( c_1 \cdot \text{fn}_t + c_2 \cdot \text{fp}_t + c_3 \cdot \text{idsw}_t \right)}{\sum_t g_t} \tag{10}$$

fpt, and where $\text{fn}_t, \text{fp}_t, \text{idsw}_t$ are the number of t the number of missed detections, the number of false detections, and the number of identity exchanges in each trajectory at

**Table 2  The experiment environment.**

| Experiment environment | Specifications |
|---|---|
| CPU | i7-11390H |
| GPUs | GTX3060 |
| IDE | Pycharm |
| Framework | Pytorch |

a time, respectively, and $g_t$ denotes the true value, and $c_1, c_2, c_3$ is a constant. To better evaluate the impact of the number of identity switches on tracking performance, *Shitrit et al. (2013)* introduced the Global Identity Switches (gidsw) metric instead of Identity Switches (idsw) to measure the number of identity switches (*De Vleeschouwer et al., 2008*).

$$\text{GMOTA} = 1 - \frac{\sum_t \left(c_1 \cdot fn_t + c_2 \cdot fp_t + c_3 \cdot gidsw_t\right)}{\sum_t g_t}. \tag{11}$$

To effectively illustrate the efficacy of KSP shortest path information and Alphapose information based on KSP in multi-target visual model training, we conducted target analysis using data solely under each modality. Building upon this, we selected two prominent methods, T-MCNF (*Ibrahim et al., 2016*) and MOT-CE (*Shitrit et al., 2013*), for comparative assessment. The T-MCNF algorithm conceptualizes the multi-target tracking problem as a multi-network streaming issue, leveraging the KSP-based methodology. This approach models the situation as a trajectory-based, multi-network flow problem, and To achieve this, the KSP algorithm (*Ghedia, Vithalani & Kothari, 2017*) is executed twice. The first iteration generates multiple trajectory segments, while the second iteration segregates these segments into distinct groups based on their appearance characteristics. Subsequently, these trajectory segments are treated as nodes to construct multiple network flow graphs, and the KSP algorithm is once again applied to derive the final trajectories. On the other hand, MOT-CE is primarily designed to track rigid body targets in three-dimensional space. This method relies on geometric position information rather than appearance characteristics for tracking.

## Experiment result and analysis

After finalizing the dataset selection and specifying the comparison methods and metrics, we proceeded with the experimental phase. The experimental settings for this article are outlined in Table 2:

On this basis, we carried out the calculation of the indicators under the three datasets, and the results in APIDIS are shown in Fig. 4.

In Fig. 4, it is evident that the proposed method attains a MOTA of 0.81 and a GMOTA of 0.64 under its evaluation indices. While not surpassing the 0.9 tracking threshold, these results are deemed acceptable, particularly in addressing the challenges posed by complex environments and multiplayer scenarios. Notably, the overall performance surpasses hybrid multi-target tracking methods at this stage. Figure 5 and 6 illustrate the test results under the volleyball and soccer datasets, respectively.

In Figs. 5 and 6, it is apparent that the model, leveraging KSP shortest path optimization and Pose information convergence as proposed, attains optimal results under both the

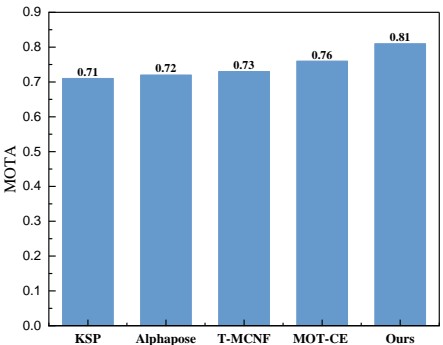
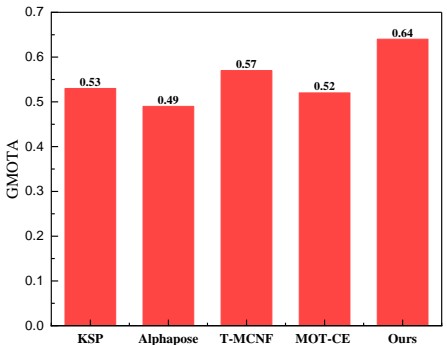

**Figure 4** The comparison result of MOTA and GMOTA on APIDIS dataset.

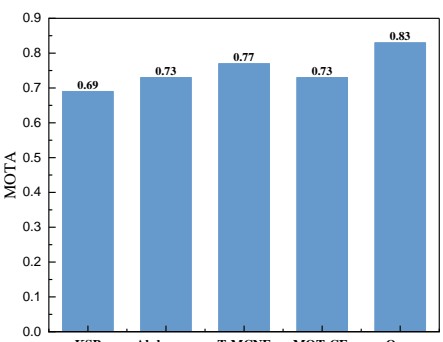
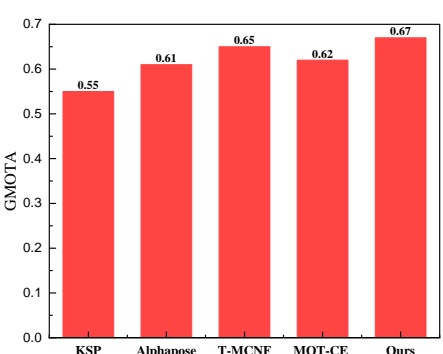

**Figure 5** The comparison result of MOTA and GMOTA on Volleyball dataset.

volleyball and soccer datasets. Specifically, the MOTA scores are 0.83 and 0.85 for the two datasets. These results outperformed the method without feature fusion and the current-stage T-MCNF and MOT-CE methods.

We computed the average metrics across the five methods using three datasets to compare model performance comprehensively. The summarized results are presented in Fig. 7.

The average results depicted in Fig. 7 demonstrate that the approach employed in this article consistently outperforms the other methods across all three datasets; this substantiates that incorporating the feature fusion method can significantly enhance the model's performance.

## The ablation experiment

Upon concluding the evaluation of model metrics across diverse datasets, we proceeded with further ablation experiments to assess the model's performance. Considering the potential impact of color variations in application scenarios, this article scrutinizes the

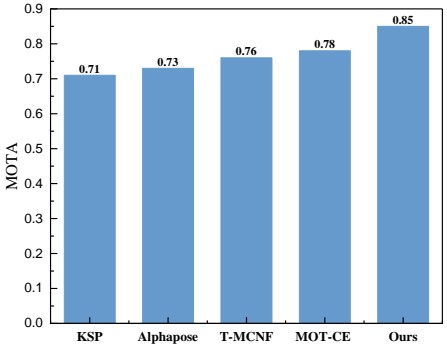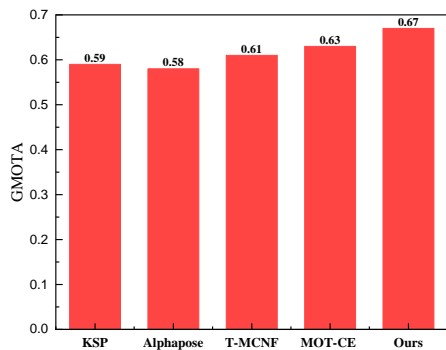

**Figure 6** The comparison result of MOTA and GMOTA on SPIROUDOME dataset.

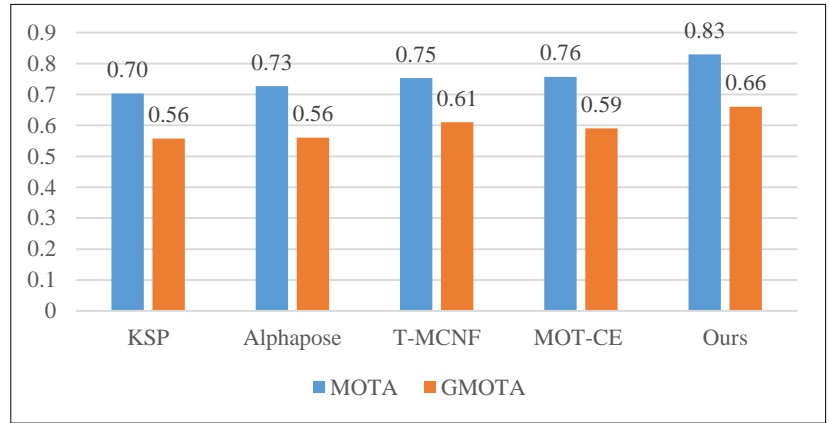

**Figure 7** The average result of MOTA and GMOTA among three datasets.

results under different color information settings. The outcomes of these experiments are illustrated in Fig. 8.

Figure 8 shows that the model's MOTA and GMOTA metrics improve when different proportions of color information are provided. Leveraging color information augmentation proves beneficial, mainly due to the distinctive color variations in the attire worn by various teams during the game. This augmentation significantly enhances the trajectory-tracking performance of the model.

Furthermore, this article compares the tracking performance between the single and convergence models under varying training iterations. The results of this comparison are presented in Fig. 9.

Figure 9 shows that the MOTA metrics notably improve after feature fusion across all three datasets. The performance is notably superior to the KSP method alone, and

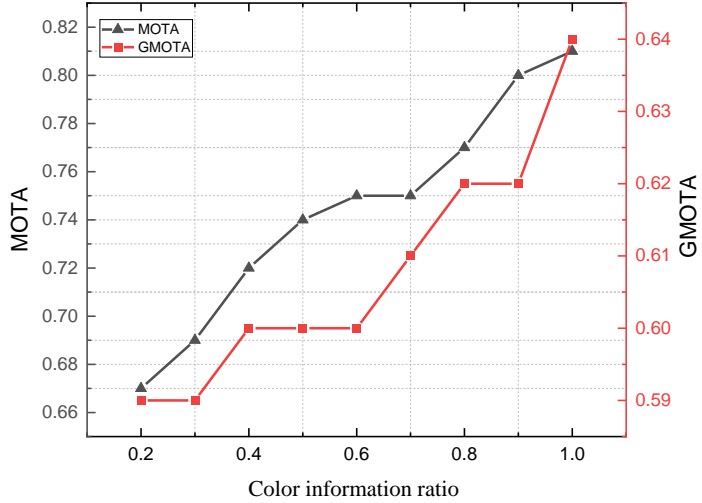

**Figure 8** The result on the APDIS datasets with different colour information ratio.

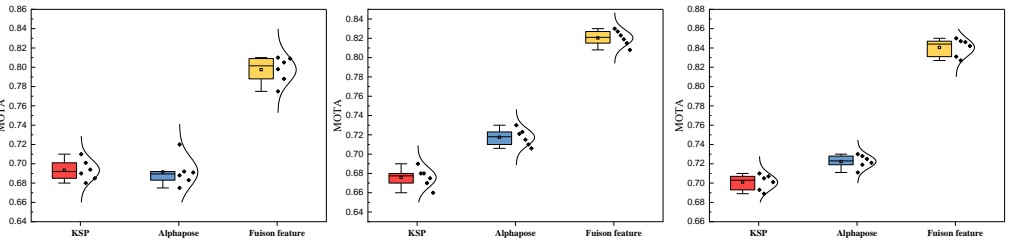

**Figure 9** The ablation experiment on the three datasets with the MOTA results.

the approach is based solely on attitude information. Additionally, when considering the overall data distribution of the model, the fusion of information enhances robustness, leading to more excellent stability in overall performance. The MOTA results showcase consistent improvement across different numbers of training iterations.

## DISCUSSION

This article delves into multi-target tracking and analysis during sports training, primarily focusing on migrating general target detection and human posture estimation algorithms from image-based applications to sports videos. The aim is to enhance algorithmic model reusability, minimize annotation costs, and reduce training expenses associated with model migration to new domains. Simultaneously, the characteristics of sports videos are leveraged to improve target detection and human pose estimation performance. Traditional feature extraction is followed by applying the KSP method for shortest path recognition, achieving path feature extraction. Subsequently, the Alphapose framework effectively completes posture estimation, facilitating identifying identity information during sports training.

Integration of sports path information and human body position information enhances multi-target recognition accuracy, surpassing the performance of traditional single KSP and alpha pose methods. Comparisons with the T-MCNF and MOT-CE methods using single image features reveal that information fusion achieves higher accuracy in target tracking, reducing computational burdens in simpler networks. Beyond target tracking, integrating deep learning with visual sensors in sports training opens avenues for real-time monitoring and evaluation of athletes' movement quality. Movement recognition and posture estimation enable coaches to gain insights into athlete performance and offer personalized guidance.

Moreover, in this article, several factors influenced the decision not to explore the experimental scenario of the Generalized Multi-Object Tracking Accuracy (GMOTA) metric under different indicator weights. Firstly, the focus of our research was to demonstrate the effectiveness of our fusion approach in enhancing multi-target tracking, primarily through improvements in robustness and accuracy. The primary aim was to validate the hypothesis that integrating KSP tracking with Alphapose could lead to superior performance in multi-target tracking tasks, particularly in sports video analysis. Combining deep learning with biosignal processing facilitates physiological state monitoring and customized training advice based on sensor information like heart rate and motion tracking.

Beyond the target tracking facilitated by image-informed visual sensors, this article envisions a broader scope of tasks achievable through target tracking in sports training— real-time monitoring and evaluation of athletes' movement quality enabled by movement recognition and posture estimation. Employing deep learning for athlete behavior analysis allows coaches to gain deeper insights into athletes' training performance and deliver personalized guidance. Integrating biosignal processing with deep learning facilitates the analysis of sensor information, including heart rate and motion tracking, to monitor athletes' physiological states. This information is then leveraged to offer customized training advice, contributing to a holistic understanding of an athlete's well-being. Combining deep learning and sensor information extends to athlete positioning, trajectory analysis, and individualized training. Video analysis and scene understanding, powered by deep learning, provide coaches with a comprehensive understanding of the overall context of a game or training session. The fusion of deep learning with multiple sensor inputs delivers thorough and accurate data analysis for sports training, empowering coaches to develop more scientific and personalized training plans tailored to each athlete's unique needs and capabilities.

## CONCLUSION

This study explored the intricacies of analyzing athlete behavior and the complexities of multi-target tracking within sports training environments. Our research introduced a novel multi-target tracking approach that integrates KSP tracking with the advanced pose estimation capabilities of Alphapose. This fusion strategy elevates the precision and reliability of tracking multiple targets in sports video data. Our empirical evaluations,

conducted across various datasets, including the challenging APIDIS, have demonstrated that this integrative model significantly enhances robustness and stability by utilizing multi-feature information fusion. By adopting a decision-level fusion strategy, we achieved noteworthy improvements in the critical performance metrics of MOTA and GMOTA, with average scores reaching 0.83 and 0.66 across the datasets examined. These scores notably surpass those achieved by methods relying on single types of information and those by other state-of-the-art approaches like T-MCNF and MOT-CE. The success of our fusion-based model in advancing MOTA and GMOTA metrics underscores its potential to significantly impact the future development of sports training systems, offering methodological insights and robust technical frameworks for enhancing multi-target tracking efficacy. In future research endeavors, we aspire to broaden the data processing modalities of the current model by incorporating multimodal sensors such as infrared and inertial sensors. This expansion aims to achieve more comprehensive data analysis and enhance the model's robustness from multiple perspectives. Additionally, we strive to extend the scope of data coverage and introduce more diverse motions for more accurate and nuanced multi-target tracking and evaluation analysis.

### Funding
This work is fully funded by the 2022-2023 scientific research project of Guangdong Provincial Sports Bureau (GDSS2022N171). The funders had no role in study design, data collection and analysis, decision to publish, or preparation of the manuscript.

### Grant Disclosures
The following grant information was disclosed by the authors:
Guangdong Provincial Sports Bureau: GDSS2022N171.

### Competing Interests
The authors declare there are no competing interests.

### Author Contributions
- Mei Hu conceived and designed the experiments, performed the experiments, performed the computation work, prepared figures and/or tables, authored or reviewed drafts of the article, and approved the final draft.
- Ming Zhang conceived and designed the experiments, performed the experiments, analyzed the data, prepared figures and/or tables, and approved the final draft.
- Kewei Yu performed the experiments, analyzed the data, performed the computation work, authored or reviewed drafts of the article, and approved the final draft.

### Data Availability
The APIDIS data is available at Zenodo: Alexander Hoelzemann, Julia Lee Romero, Marius Bock, Kristof Van Laerhoven, & Qin Lv. (2023). Hang-Time HAR: A Benchmark

Dataset for Basketball Activity Recognition using Wrist-worn Inertial Sensors [Data set]. In MDPI Sensors: Vols. Sensors 2023, 23(13), 5879 (1.0, Number Inertial Measurement Units in Sport). Zenodo. https://doi.org/10.5281/zenodo.7920485.

The Volleyball data is available at Zenodo: Mehmet Ali Arabacı, Elif Surer, & Alptekin Temizel. (2023). EOAD (Egocentric Outdoor Activity Dataset) (Version 1) [Data set]. Zenodo. https://doi.org/10.5281/zenodo.7742660

The SPIROUDOME data is available at Zenodo: Lee B. Hinkle, Gentry Atkinson, & Vangelis Metsis. (2022). TWristAR - wristband activity recognition (1.0.0) [Data set]. Zenodo. https://doi.org/10.5281/zenodo.5911808

## Supplemental Information

Supplemental information for this article can be found online at http://dx.doi.org/10.7717/peerj-cs.2030#supplemental-information.

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
