# Peer review of "Design of sports training information analysis system based on a multi-target visual model under sensor-scale spatial transformation"

_PeerJ Computer Science, doi:10.7717/peerj-cs.2030_

## Round 0.1 · original submission · Major Revisions

Dear authors,

Thank you for submitting your article. Feedback from the reviewers is now available. It is not recommended that your article be published in its current format. However, we strongly recommend that you address the issues raised by the reviewers, especially those related to readability, experimental design and validity, and resubmit your paper after making the necessary changes.

Best wishes,

**Language Note:** PeerJ staff have identified that the English language needs to be improved. When you prepare your next revision, please either (i) have a colleague who is proficient in English and familiar with the subject matter review your manuscript, or (ii) contact a professional editing service to review your manuscript. PeerJ can provide language editing services - you can contact us at [email protected] for pricing (be sure to provide your manuscript number and title). – PeerJ Staff

Reviewer 1 ·

Basic reporting

This groundbreaking manuscript delves into the contemporary landscape of athletic training, emphasizing the indispensable role of technology in elevating athlete performance and refining training outcomes. The integration of multi-target visual modeling and sensor technology creates a nuanced layer of sports training data, unveiling new dimensions in understanding athlete movements.The authors present a thought-provoking framework aimed at addressing the intricacies of multi-target tracking during sports training and competition. Leveraging the KSP methodology for shortest path determination and the Alphapose network for pose information.

Experimental design

1. When introducing the methodology, specify the significance of using the KSP methodology and Alphapose network. Provide a concise rationale for their selection.

2. Ensure a clear sequence of steps in the framework explanation to help the reader follow the process effortlessly.

3. Emphasize the significance of integrating the shortest path with pose information early on to establish a clear connection between the two components.

4. Elaborate on how the KSP methodology and Alphapose network contribute to recognizing the athlete's shortest path. Provide a brief overview of each component's role to enhance reader comprehension.

Validity of the findings

The empirical results presented in the manuscript provide compelling evidence of the efficacy of this integrated approach. The MOTA index and GMOTA index demonstrate marked improvements post-integration, surpassing those of traditional information tracking methods. This not only establishes a technical underpinning for the proposed framework but also lays a robust foundation for future advancements in information fusion within sports training analysis systems, however, following improvement needed

1. Ensure a smooth transition when describing the fusion process. Clearly delineate how the convolutional neural network and KSP converge, and subsequently, how Alphapose-derived data is integrated. This will enhance the reader's understanding of the intricate fusion process.

Additional comments

1. In the sentence discussing the amalgamation of divergent data sources, consider using a term like "convergence" instead of "fusion" for added precision in describing the process.

2. Maintain consistency in referring to Alphapose-derived pose information. Consider using the term consistently throughout to avoid potential confusion.

3. In the concluding sentences, reiterate the key findings and their implications for multi-target tracking. Clearly state how the fusion approach, grounded in Alphapose, contributes to surpassing MOTA and GMOTA index performance compared to solitary tracking methods.
Ensure the correctness of language and grammar of the whole article.

Reviewer 2 ·

Basic reporting

The authors propose an innovative multi-target tracking model that integrates KSP shortest path tracking and Alphapose pose information, aiming to optimize the efficacy of multi-target tracking in the domain of sports video analysis. To enhance the clarity and impact of this message, consider the following revisions:

Experimental design

The opening sentence effectively sets the stage but could be more specific about the type of technology and its relevance to athletic training. For instance, consider adding a brief example or context to immediately engage the reader.

 Break down the sentence discussing the framework into shorter sentences for improved readability and comprehension. For example, "Our proposed framework combines the strengths of KSP shortest path tracking and Alphapose pose information. This integration enhances the accuracy and robustness of multi-target tracking in sports videos." In addition, "In the realm of athletic training, cutting-edge technology has revolutionized how we analyze and improve athletes' performance." Moreover, Instead of "comprehensive apprehension," consider a clearer term like "comprehensive understanding" to enhance readability. "To gain a comprehensive understanding of athletes' movements, our model integrates..."

 Check the flow of the paragraph to ensure that each sentence naturally leads to the next, creating a cohesive narrative. Ensure that the progression from introducing the model to discussing specific technologies is seamless and logical. The author should make more connections between the background introduction and the adopted KSP method and the network framework of attitude information fusion.

 The author has commendably achieved the integration of decision layers pertaining to pose and path information, positioning it as the central innovation of this research endeavor. Delving into the scientific underpinnings of the referenced path information becomes imperative. Does it encompass the conventional paradigm of capturing dynamic features, or does it extend beyond to introduce novel considerations?

 In the interest of clarity and completeness, there is a suggestion to amalgamate the formulations associated with Figure 1, providing a seamless and comprehensive elucidation. Within the realm of Figure 1, the symbolic representation of each node variable 'm' denoting the current quantity of target objects at that specific node, and the attribution of each edge weight to the flow of 'f' targets, warrants further explication. Additionally, clarification is sought regarding the nature of 'N(k)'—whether it serves as a constraint function and, if so, its specific role in the formulation.

 The imperative of objectivity is paramount throughout the manuscript. As such, it is advisable to bolster Formula (6) and Formula (7) with pertinent references, thereby establishing a more robust scholarly foundation. This will assuage concerns about the originality of these formulations and align the work more closely with the existing body of literature.

 The results section, while presenting valuable findings, could benefit from a more contemporary benchmark for model comparison. The reliance on T-MCNF and MOT-CE, methods proposed over five years ago, may not fully capture the current state-of-the-art in the field. Consideration of more recent approaches would fortify the study's positioning within the evolving landscape of target tracking methodologies.

 Why was the experimental scenario of the GMOTA metric under different indicator weights not considered?

Validity of the findings

please see the above section

Additional comments

pl. improve the language of the paper.

---

## Round 0.2 · accepted · Accept

Dear authors,

Thank you for clearly addressing all the reviewers' comments. I confirm that the quality of your paper has improved. The paper is now ready for publication in light of this revision.

Best wishes,

Reviewer 1 ·

Basic reporting

This paper presents a groundbreaking approach that combines multi-target visual modeling with sensor technology to enhance the depth and quality of sports training data. I am satisfied with the revision version and its improvements.

Experimental design

The experimental design is justified enough to be considered for the manuscript.

Validity of the findings

Empirical results demonstrate the efficacy of the proposed framework in enhancing tracking accuracy, as evidenced by improvements in the Multiple Object Tracking Accuracy (MOTA) and Global Multiple Object Tracking Accuracy (GMOTA) indices. These findings underscore the technical superiority of the fusion approach over solitary information tracking methods, providing a robust foundation for future sports training analysis systems

Additional comments

overall, the quality of the article seems good to be acceptable.

Reviewer 2 ·

Basic reporting

The revised paper has been improved in light of the comments made by me earlier.

Experimental design

Overall, the paper makes substantial contributions to the field of athletic training by introducing an innovative framework, demonstrating performance improvements, and offering practical insights for future research and development. These contributions enhance the scholarly discourse and advance the state of knowledge in sports training analysis.

Validity of the findings

Empirical results indicate significant improvements in metrics such as the MOTA index and GMOTA index following the integration of information streams. This performance enhancement underscores the efficacy of the proposed framework in optimizing multi-target tracking during sports training and competition

Additional comments

I am satisfied with the revised version of the manuscript. Therefore, I recommend its acceptance.